# Dynamic of Hepatitis E Virus (HEV) Shedding in Pigs

**DOI:** 10.3390/ani12091063

**Published:** 2022-04-20

**Authors:** Giovanni Ianiro, Marina Monini, Luca De Sabato, Eleonora Chelli, Natalino Cerini, Fabio Ostanello, Ilaria Di Bartolo

**Affiliations:** 1Department of Food Safety, Nutrition and Veterinary Public Health, Istituto Superiore di Sanità, Viale Regina Elena, 299, 00161 Rome, Italy; giovanni.ianiro@iss.it (G.I.); luca.desabato@iss.it (L.D.S.); chelli.eleonora@gmail.com (E.C.); ilaria.dibartolo@iss.it (I.D.B.); 2Public Veterinary Service, Local Health Unit Roma 6, Via San Biagio, 19, 00049 Velletri, Italy; natalino.cerini@aslroma6.it; 3Department of Veterinary Medical Sciences, University of Bologna, Via Tolara di Sopra, 50, 40064 Ozzano dell’Emilia, Italy

**Keywords:** HEV, swine, pigs, fatteners, foodborne, risk, slaughterhouse, fecal shedding

## Abstract

**Simple Summary:**

Hepatitis E virus (HEV) is an emerging pathogen, causing an increasing number of autochthonous cases in industrialized countries. In Europe, infections are associated with the zoonotic HEV-3 and HEV-4 genotypes and pigs and wild boars are the main reservoirs. A major concern of infections is linked to its foodborne transmission, due to consumption of raw or undercooked pork products infected by HEV-3 or HEV-4. HEV-3 is widespread in farmed pigs, mainly aged 3–4 months. Besides a decline with age, infected pigs have been observed at slaughterhouses, representing a risk for both the consumers and the workers of the pig industry. HEV is transmitted by the fecal–oral route and shed in feces in large amounts. The risk of viral spreading in farm and presence of infected pigs at slaughtering was evaluated by assessing the quantity and the duration of HEV-3 shedding in feces of infected pigs. Feces of 23 HEV-3 positive pigs were assayed during their fattening, shortly before their slaughtering. Results confirmed a long period of viral shedding in feces with a large amount of the virus released in the environment (mean 10^5^ GC/g). Prevalence and quantity of the virus declines with the age of animals. The study provides information on the dynamic of the infection in pigs, important to prevent HEV occurrence and circulation in farms.

**Abstract:**

Genotype 3 of hepatitis E virus (HEV-3) is the most common in Europe in both humans and pigs. HEV-3 strains are zoonotic, and foodborne cases associated with consumption of raw and undercooked pork products, mainly liver sausages, have been described. HEV-3 circulates largely in European pig farms, maybe due to its long persistence in the environment. Animals get infected around 3–4 months of age; shortly after, the infection starts to decline up to the age of slaughtering (8–9 months of age in Italy). With the purpose to understand the duration in farmed pigs of the shedding of the virus and its quantity, HEV-RNA detection was performed by Real-time RT-PCR from feces collected individually from two groups of 23 pigs. Sampling was conducted for 4 months shortly before slaughtering age. At 4-months-old, all animals were shedding HEV-3 to high load around 10^5^ genome copies per gram (GC/g). Prevalence was higher in growers than in fatteners, with most of the pigs still positive around 166 days of age. Beyond some difference among individual pigs, the amount of HEV in feces decreased with the age of animals. The longest fattening period should ensure a lower risk of HEV shedder animals at slaughter, reducing the risk of food contamination.

## 1. Introduction

The hepatitis E virus (HEV) belongs to the *Hepeviridae* family which includes viruses infecting both humans and animals, being zoonotic. The species *Orthohepevirus A* includes five HEV genotypes able to infect humans: HEV-1 and HEV-2 only infect humans, while HEV-3, HEV-4 and HEV-7 are zoonotic [1,2]. In industrialized countries, over the last ten years, an increasing number of human cases have been associated with infections of HEV-3 and HEV-4 [3]. The disease is generally self-limited with mild symptoms. However, the hepatitis can become chronic in immunocompromised patients, representing a serious concern for this category of patients [4]. Both HEV-3 and HEV-4 are zoonotic, and pigs and wild boars are the main reservoirs. Other animals are also permissive to the infection such as deer, that could be infected through spill-over from the main hosts [5].

Zoonotic transmission of HEV-3 and HEV-4 is fully recognized, with nucleotide sequences of HEV-3 and HEV-4 strains obtained from humans displaying high identity with strains belonging to the same genotypes and detected in animals [6]. This finding allowed to define small clusters of cases linked to consumption of pork and wild boar derived food products, mainly containing liver [7]. In Europe, the main circulating genotype is HEV-3, while HEV-4 has been detected sporadically and is prevalent in Asia [5].

Circulation of HEV-3 in pig farms has been described worldwide. In Europe, several studies reported the detection of HEV-3 with seroprevalence varying between 30% to 98% [8]. This variability is partially due to different methodologies used, and to within-herd HEV infection dynamic [9]. However, all studies conducted in naturally infected pigs revealed that animals became infected at 8–15 weeks of age when specific maternally derived antibodies (MDA) decline [10]; the percentage of infected animals decreases in pigs aged up to 120 days, when most animals become negative and HEV-RNA is not further revealed [5,8]. This dynamic of HEV infection linked to the age of animals has also been confirmed by experimental infections of pregnant gilts, which delivered piglets with anti-HEV MDA still present at 60 days of age [11]. Older animals are generally seropositive, but a low percentage of pigs are still HEV positive in liver (2–6%) at the age of slaughtering [12,13]. Pigs still positive at the age of slaughtering represent a risk for the consumers and for swine workers, since the infection can determine the presence of HEV in the liver, in the feces and in the blood, which even though occurs rarely, can act as source of contamination of muscle, becoming a source of foodborne or occupational exposure in humans [13,14].

The duration of viremia is shorter than liver infection thus far, representing a lower risk of occurrence in animals at the abattoir. During slaughtering and dissection, the environment and utensils can also be cross-contaminated by HEV, due to contact with both feces and blood, indicating that the initial production areas (bleeding to evisceration) are at risk for cross-contamination and represent possible hazards for workers [15].

A few longitudinal studies were conducted to assess the dynamic of infection in pigs. All studies reported different age of the infection (30–90 days; 60–97 days; up to 105 days) and seroconversion, which are also influenced by MDA [16,17]. The peak of fecal shedding is reported between 3 and 4 months of age, and only a few animals are still HEV-RNA-positive at the age of 6 months [8]. The difference observed in studies confirmed that several factors may influence the occurrence of the infection in pigs, its persistence during a pig’s life and the dynamic of infection. It is proved that pigs co-infected with other pathogens (e.g., porcine reproductive and respiratory syndrome virus, PRRSV and porcine circovirus type 2, PCV2) show a longer duration of the infection [18,19].

In the present study, the duration of shedding of HEV in feces of 23 pigs from growing to the end of fattening period, shortly before their slaughter at 160 kg of body live weight (8–9 months of age) was evaluated by Real-time RT-PCR. Both qualitative and quantitative estimation of HEV in feces was conducted. Results obtained add a piece of information on the viral shedding of HEV in pigs, to understand the dynamic of infection in a two-site farrow-to-finish farm.

## 2. Materials and Methods

### 2.1. Samplings

Fecal samples were taken on a two-site farrow-to-finish farm consisting of about 500 breeders (site 1: breeding and growing production stage). Pigs are moved to the grower unit at the age of ~3 months (9-to-12-weeks-old), and then to site 2 (fattening, 50 km away) at 4 months (45 kg).

Before the study, the presence of HEV infection at site 1 was evaluated in fecal pool samples from sow and piglets. Afterward, 10 pigs (samples A01–A10) were ear-tagged and individually sampled 9 times, from growing (site 1) to fattening (site 2), every 2–3 weeks for 5 consecutive months, from the age of 110 days up to 235 days of age (batch A). After 6 months, a second group of 13 animals (samples B01–B13) was individually sampled 4 times, from the age of 90 days (from growing up to 187 days of age; batch B) (Figure 1).

Over the course of the study, two pigs lost the ear-tag after 7 samplings, one after 8 samplings and one died after the fourth sampling (all from batch A). To collect individual fecal samples, animals were temporarily moved into separate cages and feces were collected from floor.

### 2.2. RNA Extraction and Quantitative Real-Time Reverse Transcription PCR

Fecal suspensions were prepared at 10% (weight/volume) in DEPC water and 150 μL of supernatant was used for RNA extraction. Before extraction, samples were spiked with 10 μL of murine norovirus (MuNoV, 1.5 × 10^5^ TDCI_50_/mL), which was used as sample process control. Total RNA was extracted by QiampViral Mini kit (Qiagen, Monza, Italy), following manufacturer’s instructions, except for the final elution volume that was set-up to 100 μL.

MuNoV detection and calculation of recovery rate was performed as previously described [20]. Recovery rate > 5% was established as value to proceed with HEV detection. For HEV detection, 5 µL of the RNA sample was analyzed using the RNA UltraSense™ One-Step qRT-PCR System (Thermofisher Scientific, Waltham, MA, USA). For quantitative estimation of genome copies per gram (GC/g) a standard curve was built as previously described [20].

### 2.3. Statistical Analysis

The Kaplan–Meier survival curves [21] were drawn to estimate the probability of animals becoming HEV-RNA negative over the survey period. Since the animals of batch A were examined for a longer time interval, the survival time was calculated up to the age of about 185 days which corresponds to the 7th sampling for batch A (age 181 days) and to the last sampling for batch B (age 187 days). Based on the interval of time followed for samplings, the unit of time was day. The shedding time in days was calculated as the interval of time between the first HEV-detection in the individual samplings, T0 of the survey, to the time of the sampling in which feces were negative for HEV. Therefore, the non-survival of a swine was represented by the change of its status from HEV-RNA-positive to HEV-RNA-negative and survival in our case was the adverse event. Censored animals were swine that remained HEV-RNA-positive for the entire study period and pigs (n = 4, lost at follow-up) which left the study because they lost the ear-tag or died before the end of the study, after 4 (1 animal), 7 (2 animals) and 8 (1 animal) samplings. Different Kaplan–Meier survival curves were obtained for the two batches. The survival curves for these subgroups were compared using the Log-rank test to assess the effect of the grouping variable on the probability of swine becoming HEV-RNA-negative.

Statistical significance was set at *p* ≤ 0.05. All statistical analyses were performed using the software SPSS 27.0.0 (IBM SPSS Statistics, Armonk, NY, USA).

### 2.4. Sequencing

Fourteen samples positive for HEV by Real-time RT-PCR were analyzed by nested RT-PCRs amplifying a 348 bp genome fragment in the ORF2 [22]. Nucleotide sequencing of amplified genes was performed at Eurofins Genomics (Ebersberg, Germany), and the obtained sequences were uploaded into the NCBI database (https://www.ncbi.nlm.nih.gov, accessed on 15 February 2022) after editing under the accession numbers: OM818402-OM818415.

## 3. Results

Before starting the survey on pigs, the presence of HEV in the herd was investigated by testing pools from animals at different stages of the production cycle. Fifteen pools of feces were collected from 5 pens in the building housing 15–20 pregnant sow, and 5 pools from 3-weeks-old piglets. Neither sows nor the piglets were positive for HEV. Furthermore, 18 pooled fecal samples were collected from 3 barns in the building where 9-to-12-weeks-old weaners (18: six from each barn) were housed. Ten pools out of 18 were positive for HEV-RNA by Real-time RT-PCR.

After 1 week, 10 randomly selected pigs from one barn were ear-tagged and feces collected individually every 7–14 days for 4 months (batch A) (Table 1).

Six months later, following the same approach, the preliminary presence of HEV-RNA in pooled fecal samples was confirmed by revealing 8 out of 23 feces pools positive in the three barns. One week later, 13 randomly selected pigs were ear-tagged, and feces were collected individually following the same scheme described above (batch B), but pigs were followed for only 3 months (Table 1) due to COVID pandemic event.

Sequence analysis was performed from the feces of 14 randomly selected animals chosen from the first and second batch (7 from the first and 7 from the second). All HEV-RNA positive pigs were infected by the same strain belonging to the HEV-3e subtype. Overall, HEV-RNA was detected from 23 pigs collected over the course of the study. In batch A, 10 out of 10 pigs were positive on the day of the first test at the age of 110 days. In batch B, 13 out of 13 pigs were positive on the day of the first test at the age of 90 days (Table 1).

The dynamic of HEV shedding in feces by pigs of both batches was similar, as confirmed by obtaining the same probability (did not statistically differ) that animals of the two batches become HEV-RNA negative at the end the study. The logrank test statistic for the comparison of their survival curves was 1.55 (*p* = 0.21). At the end of the study, the cumulative proportion of all animals surviving as HEV-RNA positive was 0.031 (SE = 0.030). This proportion was 0.000 (SE = 0.000) in batch A and 0.046 (SE = 0.045) in batch B (Figure 2).

In the absence of precise age of infection in the investigated pigs, since pigs were positive at the first sampling, we expressed the duration of infection by referring to the animals’ ages (Figure 1). The shortest duration of virus shedding in feces was revealed in three pigs of batch A which were positive up to the age of 140 days (30–43 days from T1, Table 1) and became negative shortly after at the age of 155 days. The longest interval of shedding was in one animal that was still shedding the virus at the age of 187 (last sample HEV positive at 97 days from T1; batch B), followed by 6 animals revealing the presence HEV-RNA in feces at the age > 166 days (56 days from the T1, batch A), and being negative at the age of 181 days. In the whole group, at the age > 181 days all but one (from batch B) pigs (22 total) were negative. As expected, pigs (n = 10) tested longer were still negative at the 235 days, shortly before slaughtering (Figure 1).

The mean and median age of pigs at which animals were still shedding the virus, considering as end of shedding the age of the last HEV positivity fecal sample detected, was 143.29–140 days and 138.0–137.5 days in batch A and B, respectively (Table 2).

The Real-time qRT-PCR was used to quantify the virus shed at each stage of sampling. The median amount of virus released in the feces up to the age of 122 and 117 days, when all animals were positive, was 1 × 10^4.6^ GC/g for batch A and 1 × 10^5^ GC/g for batch B, respectively (the first two samplings for both batches revealed an approx. identical amount of virus, Table 3). The amount of HEV-RNA in feces varied among pigs, ranging of about two logs depending on sampling. The maximum amount of HEV GC/g was obtained during the first sampling, in the feces of two animals shedding 1 × 10^8^ and 1 × 10^9^ GC/g, in batch A and B, respectively (Table 3). Overall, the amount decreases with the age of the animals (Figure 3).

## 4. Discussion

Understanding the dynamic of infection of HEV in pig farms is clearly important for evaluating how the circulation of the virus could be managed.

In the present study, fecal samples neither from sow nor from piglets were HEV positive. The negative result could be partly explained by the low number of samples from breeders investigated. Conversely, the infection was significantly spread among animals at the age of 3 months after they were moved from the breeding to the growing unit, and, finally, to the finisher unit (site 2). Indeed, 48% of pooled fecal samples (growers 90–100 days old), and, after one week, 100% of the individually sampled pigs (100–110 days) tested positive for HEV, in accordance with previous studies [23,24] which estimated the age of infection between 60 and 97 days [10].

A few studies described field longitudinal studies and were mostly focused on following the antibodies response, which in turn showed an early infection (IgM) in pigs at 7 weeks of age and commonly the shedding of HEV in feces of 4-to-8-week-old pigs [25]. Pigs at slaughterhouse are mostly seropositive (50–100%) but HEV RNA was also detected in 5–9% of them [25].

Despite variability among studies, a model predicted the peak of viral shedding in feces of pigs at around the age of 90 days [8], as confirmed by our results.

In our study, pools of feces were collected from three barns, containing around 200 pigs each, which were housed in the barns 2 or 3 weeks before the sampling. HEV-RNA was retrieved in the pool of feces obtained in the three barns. The percentage of positive pools for each barn increased proportionally with the length of time of housing of animals within the barn. Subsequently, although we do not know when fecal shedding of the virus started, we can speculate that it happens after pig transfer to the growing unit and increases with the time of housing, since no positive pigs were revealed in the breeding unit. However, the number of pools analyzed from breeding is low to exclude that HEV positive pigs at breeding were missed.

During the survey, it was observed that feces contained high viral RNA loads (GC/g).

It was 10^6^ GC/g at the first samplings on growers, decreased in the next weeks but, taking into consideration both animals shedding and those not, the amount was high up to the age of 155 days (Figure 3) when a significant decrease was observed, and declined to <10^2.7^ GC/g before animals ended shedding of the virus.

The amount of HEV-RNA in feces decreases with the age of animals, which may suggest a progressive clearance of the shedding during the infection up to the negative result. Similar results have been reported previously in Japan [26]. The high prevalence of infection in the investigated farm may be explained by the quantity of virus released in the pen environment strongly influencing the probability of infection [8]. Thus far suggesting that the source of HEV may have been the environment.

This is corroborated by the amount of virus retrieved which correspond to the infectious dose estimated in pigs of 10^5^ HEV RNA copies [27]. Batches of animals are housed in the same building, at both growing and fattening stages, without any all-in all-out procedures between two successive batches. This may suggest that animals are exposed to viral particles that could persist in the environment from the previous batch of HEV positive pigs. This is also corroborated by previous findings which describe that HEV shedding is associated with movement of pigs and that mixing pigs from different batches or farms is a risk factor for a high HEV prevalence on the farm [28]. In addition, no pigs were newly introduced to the farm, reducing the possibility of an external source.

The continuous circulation of the same viral variant is confirmed by the detection of the same virus (100% nt. id.) during the whole survey and in both groups, including the second group of animals investigated from the same farm 6 months later. The long presence of the virus in the farm for up to 5 years has been largely described [29,30]. The long persistence of virus in the environment, the high quantity of virus released by feces, housing large group of animals are all risk factors contributing to the high HEV prevalence in farm [8].

We do not know the precise time when pigs started to shed the virus because at the time of first individual sampling, they were already positive. Nevertheless, the decline of virus shedding in feces was observed in 10 pigs at the age of 157–165 days old (22–23 weeks), with variable duration depending on animals. Only one pig was still shedding the virus in feces at 187 days of age. In previous studies, younger age was reported for lasting the shedding of virus in feces (21–22 weeks) [25,26] and, by modelling studies, only 6.1% of pigs still shed viruses at the age of 185 days [8].

In our study, at the age 165 days, all but one pigs were negative and kept their negativity up to the time of slaughter, that was >200 days. Furthermore, calculating the probability of pigs becoming negative during the survey, it was 3.1%, considering both batches, but decreased to 0% evaluating only batch A. It is noteworthy that this estimated probability corresponds to our previous finding. In the previous study that we conducted on pigs at the slaughterhouse, the mean prevalence was quite similar (3.6%) as revealed by HEV positivity in either feces or liver [13]. These results suggest that the long fattening period may ensure a total clearance of the shedding. In the absence of testing the liver of animals during slaughtering, we cannot rule out if they were positive, even if HEV RNA may persist in the liver after shedding and viremia have ended, and pigs are no longer considered infectious to other pigs [31]. However, the absence of HEV in feces can be a key indicator of the lower risk at slaughter for both the absence of the risk of fecal cross-contamination and the reduced probability of liver HEV positive.

One pig (second batch) was negative over the whole survey (data not shown) even sharing the same pen with positive animals. Unfortunately, since only one pig was negative and no further analyses were performed, the data cannot be further discussed but it is notable that the pig did not get infected although probably most of the pen-mates were releasing HEV by feces. The animal could be resistant to the infection, opening new hypothesis insight to the genetic of animals and their resistance to HEV.

## 5. Conclusions

In conclusion, despite the strict rules applied in pig farming, in Italy, as well as worldwide, the continued exposition of animals to HEV, due to the high resistance in the environment and to the prolonged shedding of already infected animals in their feces, highlights the need for continuative monitoring of animals to reduce the risk of foodborne transmission to humans. This surveillance should be needed at different points of the food product chain, including the stages of farming and slaughtering, together with an augmented attention to the finished products ready for the market.

## Figures and Tables

**Figure 1 animals-12-01063-f001:**
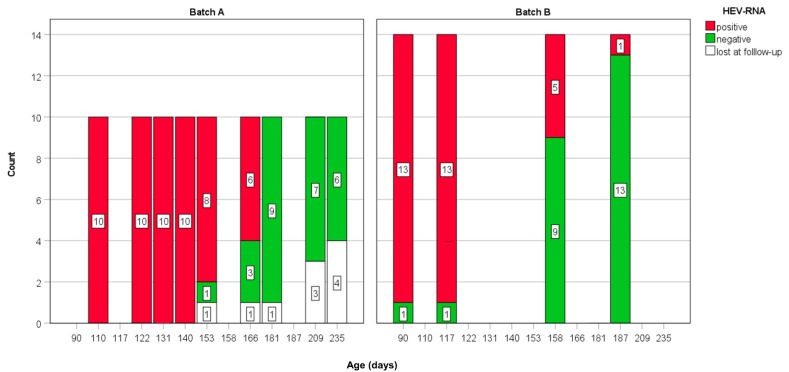
Scheme summary of HEV-RNA detection for each batch over the study period. Animals lost during the study are indicated in white (lost at follow-up because died or lost the ear-tag).

**Figure 2 animals-12-01063-f002:**
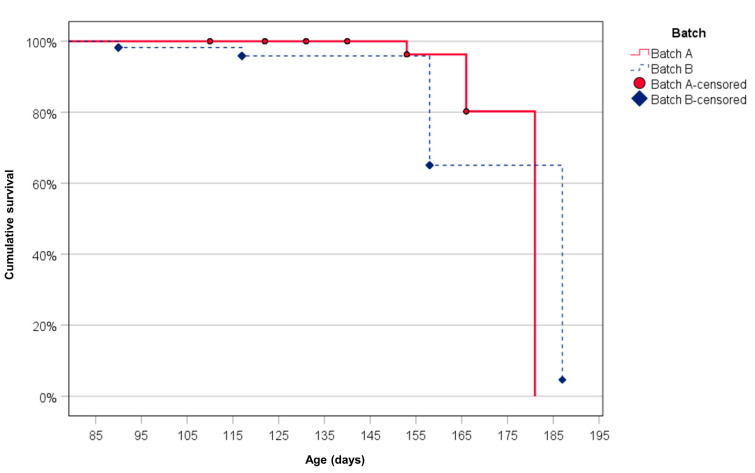
Kaplan–Meier survival curves for animals of the two batches. Censored animals are pigs remained HEV-RNA positives for the entire study period and animals of the batch A that lost the ear-tag or died (n = 4, lost at follow-up).

**Figure 3 animals-12-01063-f003:**
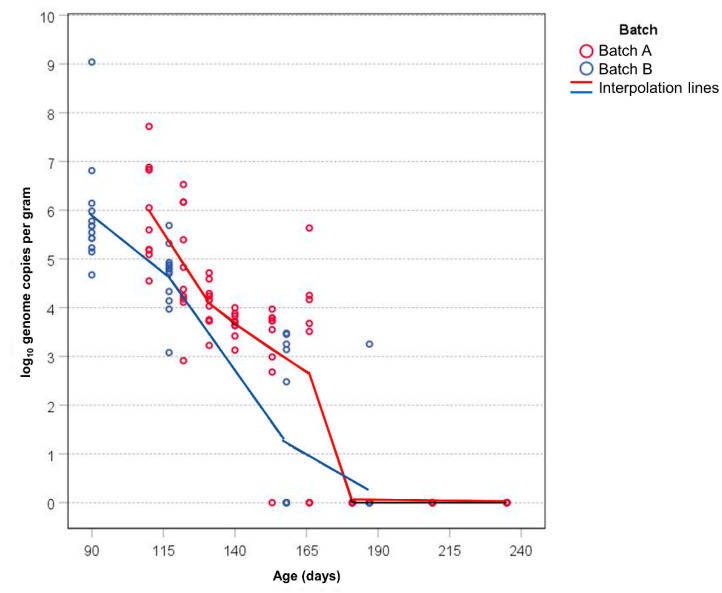
Amount of HEV-RNA expressed as log_10_ genome copies (GC) per gram of feces shed by each animal during the whole survey. Single values of GC for each pig are indicated by circle. The red colored points and line are referred to batch A, the blue to batch B.

**Table 1 animals-12-01063-t001:** Details of samples analyzed including age of animals, interval of time (days) of samplings and results of HEV-RNA detection are summarized.

Batch	Sampling (T)	Age (Days)	No. of Days from the First Sampling (T1)	HEV-RNA		
				Total Tested	Positive	Prevalence %
A	1	110	0	10	10	100
	2	122	12	10	10	100
	3	131	21	10	10	100
	4	140	30	10	10	100
	5	153	43	9	8	88.9
	6	166	56	9	6	66.7
	7	181	71	9	0	0
	8	209	99	7	0	0
	9	235	125	6	0	0
B	1	90	0	13	13	100
	2	117	27	13	13	100
	3	158	68	13	5	38.5
	4	187	97	13	1	7.7

**Table 2 animals-12-01063-t002:** Summary of characteristics of viral shedding in feces, by age of animals for 10 and 13 animals of batch A and B, respectively. The age corresponds to the last HEV positivity detected in each batch. The age of pig was considered positive up to the last sampling of HEV-RNA detection.

	Age of Animals (Days)				
Batch	Mean	Std. Dev.	Minimum	Maximum	Median
A	143.29	23.33	110	166	140.0
B	138.00	37.60	90	187	137.5

**Table 3 animals-12-01063-t003:** Summary of HEV-RNA genome copies per gram (GC/g) of feces, detected by quantitative Real-time RT-PCR at each sampling.

		Animals				GC/g ^1^			
Batch	Sampling (T)	Age (Days)	No. Tested	HEV Positive	Mean ^1^	Std. Deviation	Minimum	Maximum	Median
A	1	110	10	10	5.99	1.03	4.55	7.72	5.82
	2	122	10	10	4.89	1.15	2.91	6.53	4.60
	3	131	10	10	4.09	0.44	3.23	4.72	4.16
	4	140	10	10	3.67	0.25	3.13	4.00	3.71
	5	153	9	8	3.14	1.25	0.00	3.97	3.73
	6	166	8	6	2.66	2.29	0.00	5.64	3.59
	7	181	9	0	0.00	0.00	0.00	0.00	0.00
	8	209	7	0	0.00	0.00	0.00	0.00	0.00
	9	235	6	0	0.00	0.00	0.00	0.00	0.00
B	1	90	13	13	5.89	1.08	4.67	9.04	5.67
	2	117	13	13	4.63	0.65	3.08	5.69	4.80
	3	158	13	5	1.22	1.62	0.00	3.48	0.00
	4	187	13	1	0.25	0.90	0.00	3.25	0.00

^1^ Values indicated are expressed as the log_10_ of the GC/g obtained.

## Data Availability

Not applicable.

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
