# Peer review of "Dynamic of Hepatitis E Virus (HEV) Shedding in Pigs"

_animals, 2022, doi:10.3390/ani12091063_

Round 1
Reviewer 1 Report
comments relative to indicated rows.
17. here a 3-4 weeks age is indicated,; not consistent with rows 30, 63.
28. reports are relative to work in the swine industry (farms, slaughterhouse) even without consumption of meat, as in row 76.
47. reference ?
55. reference ?
65. RNA -ve /viral shedding negative.
80. do you mean antigent +ve ? in such a case, sentence should be amended accordingly, as in 65.
153. assuming, as in rows 62, 63, maternal antibodies are due to +ve sows, and also assuming a decrease in viral shedding after 6 months of age and, therefore also in breeders, may it be that a sample of 15 pools is not enough to detect a very low percentage of shedding animals ? a 95% confidence for a 5% prevalence from 5 sows x 20 boxes = 100 sows, should be something around 40 samples/pools
219, 220 as above for 153.
Sows and piglets' (younger than 90 days) shedding data looks problematic; in other works, piglets 1,5 months of age resulted already sero+ve and already RNA +ve at 2,5 months of age. Early serological positivity can be reconducted to sows positivity and, therefore, also a certain degree of RNA +ve breeders could be expected.
If these data are not available, as we understand from the paper, at least RNA -ve status of breeders should be explained (insufficient sampling ?) AND integrated with literature references/comments about supposed source of infection of piglets.
Author Response
Reviewer: 1
Comments and Suggestions for Authors
Comments relative to indicated rows
- here a 3-4 weeks age is indicated,; not consistent with rows 30, 63.
Reply: The typo was changed
- reports are relative to work in the swine industry (farms, slaughterhouse) even without consumption of meat, as in row 76.
Reply: we added the information suggested and the sentence was changed accordingly.
- reference ?
Reply: We added a reference to support the statement
- 55. reference ?
Reply: We added a reference to support the statement
- 65. RNA -ve /viral shedding negative.
Reply: The sentence was modified accordingly; the cited papers reported HEV-negative referred to RNA.
- 80. do you mean antigent +ve ? in such a case, sentence should be amended accordingly, as in 65.
Reply: We mean HEV-RNA detection. The sentence was modified accordingly.
- assuming, as in rows 62, 63, maternal antibodies are due to +ve sows, and also assuming a decrease in viral shedding after 6 months of age and, therefore also in breeders, may it be that a sample of 15 pools is not enough to detect a very low percentage of shedding animals ? a 95% confidence for a 5% prevalence from 5 sows x 20 boxes = 100 sows, should be something around 40 samples/pools
219, 220 as above for 153.
Sows and piglets' (younger than 90 days) shedding data looks problematic; in other works, piglets 1,5 months of age resulted already sero+ve and already RNA +ve at 2,5 months of age. Early serological positivity can be reconducted to sows positivity and, therefore a certain degree of RNA +ve breeders could be expected.
If these data are not available, as we understand from the paper, at least RNA -ve status of breeders should be explained (insufficient sampling?), and integrated with literature references/comments about supposed source of infection of piglets.
Reply: we agree with the reviewer. We cannot rule out that insufficient sampling could have determine the negative results and that sow are also a source of infection. However, the investigated sows, resulted negative for HEV-RNA, were older than 6-7 months, no gilts were housed in the investigated barns, suggesting that the negative status of breeders could be linked to the older age of pigs with a less probability to still be infected (Refs. Ianiro et al., 2021; Withenshaw et al., 2022) A sentence explaining the limitations highlighted by this reviewer and possible explanation of RNA negativity observed in sow was added at line 231.
Reviewer 2 Report
In the present manuscript, the authors have studied Hepatitis E virus shedding in pigs in two-site farrow-to-finish farms. The manuscript is of interest for the field as it gives evidence that long fattening period are needed to lower the risk of pigs being HEV positive at slaughter. However, several points should be clarified to improve the quality of the manuscript as discussed below.
Major points:
- The study could have benefited from higher number of pigs and time points sampled to better analyse the dynamics of HEV shedding (it was unfortunately not possible in part because of the COVID pandemic). Moreover, as pigs were all positive at the first time point of sampling, it was unfortunately not possible to make any analysis on the duration of HEV shedding in faeces.
- Figure 2: I find it difficult to follow the relevance and need of the statistical analysis performed on this figure. Can you really compare the dynamic of HEV shedding of both batches as the time points of sampling used are not the same (line 173).
- Table 2 represents the mean age at which pigs were found HEV-positive for the last time. This should be better explained in the result section (line 191-192) and table legend (line 195-197).
- Moderate English changes are needed and some sentences should be rephrased as the text is sometimes difficult to follow.
Minor points:
- Line 17: 3-4 weeks or months?
- Line 14, 28, 47: associated with
- Line 60 : seroprevalence rather than prevalence
- Line 64: with age up to ?
- Line 69-76: sentences not very clear
- Line 89: quantitative rather than quantitate
- Figure 1: I find this figure confusing regarding the red/green indicators, especially for the second batch. As sampling was performed up to 40 days apart, it is not very accurate to show that the pig was either positive or negative all the time between the time points of sampling (at least between a negative and a positive result).
- If I understood well, HEV positive samples were found in site 1 but only in the grower unit and not the breeding unit and individual sampling started before the pigs were moved to site 2. When were the pigs moved to site 2?
- Figure 2: Are censored pigs the one becoming negative for HEV-RNA (figure legend) or the ones remaining positive (as written in material and method)? In the figure legend (Batch A/B-censored), you mean that the censored pigs were included or not?
- Line 206: How did you determine the median amount of virus released at the age of 120 days?
- Figure 3: It would be easier to read the figure if you draw the interpolation lines in red and blue according to the batch represented or if you define both in the legend.
- Line 235: What do you mean by a prevalence as high as the length of time of housing?
- Line 238: sentence not clear
- Line 285: Why has this negative animal been removed? It could appear at least in figure 1 (even if not included in further analysis). It is interesting data. Is this animal resistant? Or was the animal infected previously for a short period?
Author Response
Response to Reviewer 2
Comments and Suggestions for Authors
In the present manuscript, the authors have studied Hepatitis E virus shedding in pigs in two-site farrow-to-finish farms. The manuscript is of interest for the field as it gives evidence that long fattening period are needed to lower the risk of pigs being HEV positive at slaughter. However, several points should be clarified to improve the quality of the manuscript as discussed below.
Reply: we acknowledge the reviewer for the useful suggestions provided. We implemented the manuscript following the suggestions and comments by the reviewer.
Major points:
- The study could have benefited from higher number of pigs and time points sampled to better analyse the dynamics of HEV shedding (it was unfortunately not possible in part because of the COVID pandemic). Moreover, as pigs were all positive at the first time point of sampling, it was unfortunately not possible to make any analysis on the duration of HEV shedding in faeces.
Reply: We are planning further studies to better study the early stage of infection.
- Figure 2: I find it difficult to follow the relevance and need of the statistical analysis performed on this figure. Can you really compare the dynamic of HEV shedding of both batches as the time points of sampling used are not the same (line 173).
Reply: we aimed to compare by statistical analyses the duration of viral shedding in the two groups, if besides the time points of sampling the two groups were comparable sharing the same environment and being moved within the farm at the same age. To overcome the limit of the time points of sampling, as properly suggested by the rev, we used the censored curve by Kaplan-Meier curves which offer a way of dealing with differing survival times without the issues of time of samplings
- Table 2 represents the mean age at which pigs were found HEV-positive for the last time. This should be better explained in the result section (line 191-192) and table legend (line 195-197).
Reply: yes the reviewer is correct. To make it clearer, both the result section (line 196-198) and table legend were modified accordingly.
- Moderate English changes are needed, and some sentences should be rephrased as the text is sometimes difficult to follow.
Reply: an English language revision was performed throughout the manuscript.
Minor points:
- Line 17: 3-4 weeks or months?
Reply: The typo was changed
- Line 14, 28, 47: associated with
Reply: The typos were changed
- Line 60: seroprevalence rather than prevalence
Reply: Corrected accordingly
- Line 64: with age up to ?
Reply: The sentence was modified accordingly
- Line 69-76: sentences not very clear
Reply: sentences were modified to make them clearer
- Line 89: quantitative rather than quantitate
Reply: The typo was changed
- Figure 1: I find this figure confusing regarding the red/green indicators, especially for the second batch. As sampling was performed up to 40 days apart, it is not very accurate to show that the pig was either positive or negative all the time between the time points of sampling (at least between a negative and a positive result).
Reply: we accepted the reviewer suggestion and changed the figure. We hope that the new figure is clearer since the time between “time points of samplings” is not further indicated to avoid improper assumption as suggested by the rev.
- If I understood well, HEV positive samples were found in site 1 but only in the grower unit and not the breeding unit and individual sampling started before the pigs were moved to site 2. When were the pigs moved to site 2?
Reply: yes, the reviewer is correct. We described it in the par 2.1 Sampling. We investigated in site 1 breeders, that were HEV-RNA negative and young growers that were positive. We started individual sampling in site 1 following growers to site 2 (moved at the age of around 4 months).
- Figure 2: Are censored pigs the one becoming negative for HEV-RNA (figure legend) or the ones remaining positive (as written in material and method)? In the figure legend (Batch A/B-censored), you mean that the censored pigs were included or not?
Reply: we apologize for the confusion, there was a mistake. The censored are animals that do not continue in the study because died or were lost (only from batch A) and those animals remaining positive. We changed the definition over the whole manuscript.
- Line 206: How did you determine the median amount of virus released at the age of 120 days?
Reply: the median amount of virus released was calculated on the set of GC per gram (using the standard curve described by De Sabato et al.,) for each pig. At the age of 120 days, 122 and 117 days for batch A and B, respectively, animals of both batches were all positive. The median HEV RNA Genome Copies per gram at 120 days was calculated as follows: the Ct obtained with the RT-qPCR were converted in GC/g based on the standard curve described by De Sabato et al. 2019, considering the amount of fecal suspension (150uL) subjected to nucleic acid extraction. The GC/g, are reported in table 3 as the LOG10 of the GC obtained, the median amount was calculated as the median LOG10 obtained and then reconverted in GC/g. A sentence has been added as an integration of paragraph 2.2 of the Materials&Methods section (lines 129-130).
- Figure 3: It would be easier to read the figure if you draw the interpolation lines in red and blue according to the batch represented or if you define both in the legend.
Reply: the interpolation lines have been coloured and described in the legend accordingly.
- Line 235: What do you mean by a prevalence as high as the length of time of housing?
Reply: We agree with the reviewer and have revised the sentence and add details to make it clearer (lines 247-254).
- Line 238: sentence not clear
Reply: The sentence was retyped to make it clearer
- Line 285: Why has this negative animal been removed? It could appear at least in figure 1 (even if not included in further analysis). It is interesting data. Is this animal resistant? Or was the animal infected previously for a short period?
Reply: we fully agree with the reviewer about the interest of result. We considered surprising that the animal was negative during the whole survey; therefore, we mentioned the result in the discussion to make aware the scientist of this possibility. With the result, both resistance to the infection or very short shedding could be hypothesized and object of future study. However, this result is not significant in the framework of our analyses, since it is based on only one animal, therefore it was excluded in further analyses to avoid adding data not supported by number. Following your consideration we added a sentence on this point.
Round 2
Reviewer 1 Report
Comments according to rows:
19: previous comment about row 28 was solved at row 19; a reference is needed.
80: still not clear; in your reply you wrote:
"Reply: We mean HEV-RNA detection. The sentence was modified accordingly".
But in facts, in the text, it is written: "anti-HEV IgG-positive....." which is referring to antibody titer., and not to viral RNA detection. The sentence should be corrected, skipping the refence to IgG.
Unless you mean two diffente concepts: peak of viral shedding at 3-4 months, AND/BUT still some IgG positivities at 6 months of age; but, in such a case, other works refer about high % of IgG positive pigs at 6 months of age and older.
But, at best of my understanding, this work is dealing with viral shedding, and not with a serological invedstigation; therefore the sentence referring to "anti-HEV IgG" is not consistent.
Author Response
Comments according to rows:
19: previous comment about row 28 was solved at row 19; a reference is needed.
Answer: We agree with the reviewer's comment. However, the Journal does not allow references to be included in the abstract. The risk for the consumer and for workers, related to viraemic pigs at the slaughterhouse is described in the sentence (which has been modified) on lines 74-78 and two references [13,14] are included.
80: still not clear; in your reply you wrote:
"Reply: We mean HEV-RNA detection. The sentence was modified accordingly".
But in facts, in the text, it is written: "anti-HEV IgG-positive....." which is referring to antibody titer, and not to viral RNA detection. The sentence should be corrected, skipping the refence to IgG.
Unless you mean two diffente concepts: peak of viral shedding at 3-4 months, AND/BUT still some IgG positivities at 6 months of age; but, in such a case, other works refer about high % of IgG positive pigs at 6 months of age and older.
But, at best of my understanding, this work is dealing with viral shedding, and not with a serological invedstigation; therefore the sentence referring to "anti-HEV IgG" is not consistent.
Answer: We apologize for the mistake, we agree with the reviewer's comment, we wrongly reported anti-HEV IgG. The sentence was modified accordingly (lines 86-88).
“The peak of fecal shedding is reported between 3 and 4 months of age and only a few animals are still HEV-RNA positive at the age of 6 months”.